# Physiological Noise Filtering in Functional Near-Infrared Spectroscopy Signals Using Wavelet Transform and Long-Short Term Memory Networks

**DOI:** 10.3390/bioengineering10060685

**Published:** 2023-06-04

**Authors:** So-Hyeon Yoo, Guanghao Huang, Keum-Shik Hong

**Affiliations:** 1School of Mechanical Engineering, Pusan National University, 2 Busandaehak-ro, Geumjeong-gu, Busan 46241, Republic of Korea; bsh00156@pusan.ac.kr; 2Institute for Future, School of Automation, Qingdao University, Qingdao 266071, China; hgh@qdu.edu.cn

**Keywords:** functional near-infrared spectroscopy, filtering, physiological noise, maximal overlap discrete wavelet transform, long-short term memory

## Abstract

Activated channels of functional near-infrared spectroscopy are typically identified using the desired hemodynamic response function (dHRF) generated by a trial period. However, this approach is not possible for an unknown trial period. In this paper, an innovative method not using the dHRF is proposed, which extracts fluctuating signals during the resting state using maximal overlap discrete wavelet transform, identifies low-frequency wavelets corresponding to physiological noise, trains them using long-short term memory networks, and predicts/subtracts them during the task session. The motivation for prediction is to maintain the phase information of physiological noise at the start time of a task, which is possible because the signal is extended from the resting state to the task session. This technique decomposes the resting state data into nine wavelets and uses the fifth to ninth wavelets for learning and prediction. In the eighth wavelet, the prediction error difference between the with and without dHRF from the 15-s prediction window appeared to be the largest. Considering the difficulty in removing physiological noise when the activation period is near the physiological noise, the proposed method can be an alternative solution when the conventional method is not applicable. In passive brain-computer interfaces, estimating the brain signal starting time is necessary.

## 1. Introduction

In processing functional near-infrared spectroscopy (fNIRS) signals, a task-related hemodynamic signal cannot be identified if a physiological noise period is overlapped with the designed task period. This study proposes a novel method to identify physiological noises from the resting state and remove those noises during the task period using wavelet techniques and neural networks-based prediction. FNIRS is a brain-imaging technique that uses two or more wavelengths of light in near-infrared bands to measure changes in oxidized and deoxidized hemoglobin concentration in the cerebral cortex [1]. When a person moves, thinks, or receives an external stimulus, the nerve cells in cerebral cortical layers become excited. As the cells require more energy, the oxidized hemoglobin concentration around the nerve cells increases, and the deoxygenated hemoglobin concentration decreases [1]. Based on this principle, fNIRS can measure brain activity in real time. Because fNIRS is inexpensive, easy to use, and harmless to the human body, it has been used in brain disease diagnosis [2,3], brain-computer interface (BCI) [4], decoding sensory signals [5,6], child development [7], and psychology research [8].

An fNIRS channel consists of one source and one detector. When the light is emitted from a light source, photons pass through several layers, including the scalp, skull, cerebrospinal fluid, capillaries, and cerebral cortex, before returning to a detector. Through this process, the detected light contains various noises that make it challenging to know the hemodynamic responses. These noises include heartbeat, breathing, and motion artifacts [9]; more problematically, very low-frequency noise around 0.01 Hz has been reported [10,11,12]. 

In improving the accuracy of the measured signal, noise removal/reduction techniques are indispensable. Various techniques can remove physiological noises such as heartbeat, breathing, and Mayer waves. For instance, the superficial noise in the scalp can be removed using short separated channels [13], additional external devices, or applying denoising techniques such as adaptive filtering [14] and correlation analysis methods [15]. In addition, since the frequency bands of physiological noise are roughly known, a band-pass filter has become one of the most easily applied noise reduction techniques [16].

A general linear model (GLM) method has been widely used to find the task-related hemodynamic response in the fNIRS signal after preprocessing [17]. The desired hemodynamic response function (dHRF), which should be used for the GLM method, is designed considering the experimental paradigm. However, in the case that the essential frequency of the dHRF overlaps with a specific frequency of physiological noise, the conventional GLM method will not work and may result in mistaking noise for the hemodynamic response. Therefore, a new different denoising technique must be pursued.

A discrete wavelet transform (DWT) is a mathematical tool used to analyze signals in the time-frequency domain [18]. In fNIRS research, DWT has been used for denoising [19,20] and connectivity analysis [21,22]. The maximal overlap discrete wavelet transform (MODWT) is a type of DWT often used in signal processing and time series analysis [23]. It decomposes a signal into a series of wavelet coefficients at different widths and time locations. Unlike the usual DWTs, which use non-overlapping sub-signal windows to perform the wavelet decomposition, the MODWT uses overlapping sub-signal windows. This nested-window approach allows the MODWT to improve time-frequency localization and reduce the boundary effects that can occur in DWTs [24]. Due to this advantage, MODWT has been applied to a wide range of signals, including audio signals [25], weather information [26,27], and biomedical signals [28]. MODWT is powerful when the signals are abnormal or have complex frequency components.

Deep learning, a subfield of artificial intelligence, is based on artificial neural networks. In recent years, brain research has increasingly used it to analyze large, complex data sets, such as those generated by biomedical devices [29]. Research has been conducted to analyze health data such as magnetic resonance imaging (MRI) [30], electrocardiograms (ECG) [31] and electroencephalograms (EEG) [32,33], or to decode brain waves to control BCI [34,35]. Furthermore, analyzing brain neuroimaging data and identifying patterns associated with specific diseases can help with early diagnosis and personalized treatment.

Long short-term memory (LSTM) is a type of recurrent neural network (RNN) architecture designed to overcome the limitations of traditional RNNs in handling long-term dependencies in sequential data [36]. It has been used in a wide range of applications for time-series data classification and forecasting [37,38,39,40]. LSTMs are particularly useful in tasks that require modeling long-term dependencies in sequential data. LSTMs’ ability to selectively remember and forget information over time is vital for accurate forecasting.

MODWT-LSTM-based prediction research has shown excellent results in predicting periodic data such as water level [41], ammonia nitrogen [42], weather [43], etc. In brain research, MODWT has been applied as a preprocessing method for EEG-based seizure detection [28,44], Alzheimer’s diagnosis [45], and resting state network analysis of fMRI [46]. Since brain signals are measured in time series, active research on brain signal classification [47,48] uses LSTM. However, to our knowledge, this is the first study to predict the noise in fNIRS signals despite many of the noise components being periodic. 

In this study, one thousand synthetic data are generated, assuming 600-s rest and 40-s task. Each data is decomposed into eight levels by the MODWT. Five wavelets containing low-frequency components from the 600-s data are used to train an LSTM network. The trained LSTM networks are used to predict the next 40 s, presumably the predicted signals of the low-frequency oscillations. The predicted signals are then subtracted from the task period data. For validation purposes, the predicted signal and original data are compared by calculating mean absolute errors (MAEs), and root mean square errors (RMSEs). Finally, the proposed method is demonstrated by analyzing the actual fNIRS data from humans.

This paper is organized as follows: Section 2 describes the proposed method on the synthetic data, Section 3 demonstrates the proposed method with actual fNIRS data, Section 4 discusses the results of this study and its applications, and Section 5 presents conclusions.

## 2. Method Development

This section describes the development of the proposed method with the following four subsections. In the first subsection, the method of synthetic fNIRS data generation is described. The second and third subsections explain the operation of MODWT and LSTM, respectively. The fourth subsection describes the validation of the proposed method. The last subsection presents the results of the data analysis.

### 2.1. Synthetic fNIRS Data Generation

One thousand synthetic data are generated according to the method of Germignani et al. [49] with a sampling frequency of 8.138 Hz. For each data, thirty orders of autoregressive noise are added to the baseline noise [50]. The synthetic physiological noises include frequency ranges of 1 ± 0.1 Hz, 0.25 ± 0.01 Hz, and 0.1 ± 0.01 Hz for cardiac, respiratory, and Mayer waves, respectively. In addition, a sine wave with a frequency of 0.01 ± 0.001 Hz was generated for the very low-frequency component [11]. The amplitudes of five signals in a synthetic fNIRS signal were set randomly in the range of 0.01 to 0.03. In this paper, the resting period is set to 10 min, considering that the concerned low-frequency noise is near 0.01 Hz.

For five hundred data samples only, the desired hemodynamic function (dHRF) based on a 2-gamma function with 20 s of task and 20 s of resting state after the 10 min resting state were added. The amplitude of this signal was randomized between 0.1 and 0.35 and added to the previously generated noise. All data were set to zero at the starting point before processing the signals. Figure 1 depicts synthetic signals for various noises and the resultant HbO signal assumed. 

### 2.2. Maximal Overlap Discrete Wavelet Transform

The discrete wavelet transform (DWT) is a signal processing technique that decomposes a signal into different frequency components at multiple levels of resolution. The DWT works by convolving the signal with a set of filters, called wavelet filters, which capture different frequency bands. The signal is decomposed into approximation and detail coefficients [19], which represent low-frequency components and high-frequency components, respectively. This decomposition is applied recursively to the approximation coefficients to obtain a multi-resolution representation. However, the DWT has several drawbacks, including the introduction of boundary artifacts due to the filtering process, the lack of shift invariance in the decomposition, and the potential loss of fine detail at higher decomposition levels. 

Zhang et al. (2018) [51] utilized the DWT in forecasting vehicle emissions and specifically compared four cases: The autoregressive integrated moving average (ARIMA) model, LSTM, DWT-ARIMA, and DWT-LSTM. They reported that adopting DWT improved the performance overall. Individually, between ARIMA and LSTM, LSTM performed better; between ARIMA and DWT-ARIMA, DWT-ARIMA generated improved results; between LSTM and DWT-LSTM, DWT-LSTM was superior; and between DWT-LSTM and DWT-ARIMA, DWT-LSTM demonstrated the best forecasting.

MODWT is a mathematical technique that transforms a signal into a multilevel wavelet and scaling factor. MODWT has several advantages over DWT. For example, the MODWT can be adequately defined for signals of arbitrary length, whereas the DWT is only for signals of integer length to the power of two.

For discrete signal $X=\{X_{t},t=0,1,\cdots ,n-1\}$, the *j*th element $W_{j}$ and scaling factor $V_{j}$ of the MODWT are defined as follows.
(1)$$W_{j,t}=\sum_{l=0}^{n-1}\overset{\sim \circ }{h_{j,l}}X_{t-l\mathrm{mod}n},\;j=1,2,\cdots ,L,$$
(2)$$V_{j,t}=\sum_{l=0}^{n-1}\overset{\sim \circ }{g_{j,l}}X_{t-l\mathrm{mod}n},$$
where $W_{j,t}$ is the wavelet coefficient of the *t*th element of the *j*th level of the MODWT; $V_{j,t}$ is the scaling factor of the *t*th element of the *j*th level; $\overset{\sim \circ }{h_{j,l}}$ and $\overset{\sim \circ }{g_{j,l}}$ are the *j*th level’s high- and low-pass filters (wavelet and scaling filters) of MODWT generated by periodizing $\tilde{h_{j,l}}$ and $\tilde{g_{j,l}}$, respectively, with *n* lengths; $\tilde{h_{j,l}}$ and $\tilde{g_{j,l}}$ are the *j*th level MODWT high ($\tilde{h_{j,l}}\equiv h_{j,l}/2^{\frac{j}{2}}$) and low ($\tilde{g_{j,l}}\equiv g_{j,l}/2^{\frac{j}{2}}$) pass filters; $h_{j,l}$ and $g_{j,l}$ are the *j*th level DWT high-pass and low-pass filters, where *L* is the maximum decomposition level. The filters are determined by the mother wavelet as in the DWT [52]. The MODWT based multiresolution analysis is expressed as follows.
(3)$$X=\sum_{j=1}^{L}D_{j}+A_{J0},$$
(4)$$D_{j,t}=\sum_{l=0}^{n-1}\overset{\sim \circ }{h_{j,l}}W_{j,t+l\mathrm{mod}n},$$
(5)$$A_{j,t}=\sum_{l=0}^{n-1}\overset{\sim \circ }{g_{j,l}}V_{j,t+l\mathrm{mod}n},$$
where $A_{L}$ is the approximation component and $D_{j}$ is the detail components (j=1,2,⋯,L). Figure 2 shows a scheme of MODWT-based multiresolution analysis.

In this study, Sym4 was selected as the mother wavelet because it resembles the canonical hemodynamic response function. Let the number of data be *N*. Then, the maximum decomposition level becomes less than log_2_(*N*). Considering our case’s shortest resting state of 60 s, the data size is 60 s × 8.13 Hz = 487.8. Therefore, the decomposition level in our work was selected by 8, which is the largest integer less than log_2_(487.8). The eight decompositions result in nine signals, of which only five signals belonging to low frequencies will be predicted.

### 2.3. Long Short-Term Memory

LSTM is a type of RNN architecture that addresses the vanishing gradient problem and allows for capturing long-term dependencies in sequential data. LSTM consists of memory cells that store and update information over time. The primary function of an LSTM is to use memory cells that can hold information for long periods. Memory cells can selectively forget or remember information based on input data and past states. This allows the network to learn and remember important information while ignoring irrelevant or redundant information. An LSTM network has three gates (input gate, forget gate, and output gate) that control the flow of information into and out of the memory cells. The input gate i(t) determines which information is stored in the memory cell c(t), the forget gate f(t) determines which information is discarded, and the output gate o(t) controls the output of the memory cell (Figure 3) [53].

The LSTM model is represented by the following equations:(6)$$a(t)=\sigma (W_{i}x(t)+U_{i}h(t-1)+b_{i}),$$
(7)$$f(t)=\sigma (W_{f}x(t)+U_{f}h(t-1)+b_{f}),$$
(8)$$\tilde{c}(t)=\tanh(W_{c}x(t)+U_{c}h(t-1)+b_{c}),$$
(9)$$c(t)=f_{t}\times c(t-1)+i_{t}\times \tilde{c}(t),$$
(10)$$o(t)=\sigma (W_{o}x(t)+U_{o}h(t-1)+b_{o}),$$
(11)h(t)=o(t)×tanh(c(t)),
where c(t−1) and c(t) are the cell states at *t* − 1 and *t*, and at each gate, $b_{i},b_{f},b_{c},b_{0}$ are the bias vectors, $W_{i},W_{f},W_{c},W_{0}$ are the weight matrices, and $U_{i},U_{f},U_{c},U_{0}$ are the recurrent weights. σ is a sigmoid function, tanh is a hyperbolic tangent activation function, and × denotes the cross product of two vectors.

In this study, three LSTM layers were utilized, with the number of hidden units set to [128, 64, 32], and a dropout layer was employed between the LSTM layers with a probability of 0.2 to prevent overfitting (Figure 4). To train the LSTM network, the Adam optimizer was used with a maximum epoch of 100 and a minibatch size of 128. All data were normalized before training.

For the synthetic data, nine hundred data were randomly selected from the thousand data to train the network, and then one hundred data were tested. The number of data points trained was divided into five conditions ([600 s, 300 s, 150 s, 90 s, 60 s] × sampling rate (8.13 Hz) = [4883, 2441, 1221, 732, 488]), and then 244 data points (30 s × 8.13 Hz) were predicted.

For actual fNIRS data, a leave-one-out method was used to avoid splitting data from the same person for training and testing. For example, to train an LSTM network to predict the 48 channels of a subject, a total of 432 channels (nine subjects × 48 channels) were used. Since the data was only 600 s long, 570 s of data were used for training, and the trained LSTM network predicted the next 30 s.

### 2.4. Validation

To determine the accuracy of the signal predicted by the LSTM, the mean absolute error (MAE) and root mean squared error (RMSE) were calculated compared to the original signal, divided by the signal with and without dHRF. The data was segmented, analyzed, and predicted to find the required resting-state length to achieve optimal prediction accuracy, as shown in Figure 5. MAE and RMSE can be calculated using the following equations.
(12)$$\mathrm{MAE}=\frac{1}{n}\sum_{i=1}^{n}\left|y_{i}-{\hat{y}}_{i}\right|,$$
(13)$$\mathrm{RMSE}=\sqrt{\frac{\sum_{i=1}^{n}{\left(y_{i}-{\hat{y}}_{i}\right)}^{2}}{n}},$$
where $y_{i}$ is the original signal, ${\hat{y}}_{i}$ is the predicted signal, i is the timestep, and n is the number of data. The calculated MAEs and RMSEs of the signal with and without the dHRF were compared using a two-sample *t*-test. 

### 2.5. Synthetic Data Analysis

The synthetic data were decomposed into nine components using MODWT, and the components used for prediction were the fifth through ninth. The frequency of the fifth wavelet was between 0.13 and 0.26 Hz, the sixth between 0.067 and 0.13 Hz, the seventh between 0.035 and 0.067 Hz, the eighth between 0.017 and 0.035 Hz, and the ninth consisted of signals below 0.017 Hz. Figure 6 shows the prediction results of the signal with and without dHRF. The signal with dHRF showed a significant fluctuation during the task period in the low-frequency signals of Wavelets 6–9, and the predicted signal did not follow this fluctuation. 

Figure 7 and Table 1 show the calculated MAEs and RMSEs. In all conditions, the MAEs and RMSEs of the signal with dHRF corresponding to Wavelets 6–9 and the signal without dHRF were statistically significantly different. The only statistically significant difference between with and without dHRF was found in the RMSE of Wavelet 5 when the MODWT-LSTM analysis was performed with 300 s of data (Figure 7c). To compare the prediction results for each condition, MAEs and RMSEs for all conditions are shown in Figure 8. The error of the dHRF signal was the largest in Condition 2 (MODWT-LSTM at 600 s) and the smallest in Condition 6 (MODWT-LSTM with 60 s of data). In particular, the difference in prediction accuracy between with and without dHRF signals of Wavelet 8 was the largest in all conditions.

For the 600 s data prediction results, MAEs and RMSEs were calculated for 1 s, 3 s, 5 s, 10 s, 15 s, and 30 s (Figure 9). In all cases, there were statistically significant differences in Wavelets 6–9 between with and without dHRF. Especially for Wavelet 7, with the most significant difference at 1 s and a decrease after that, but for Wavelet 8, the difference started at 10 s and was most extensive at 15 s.

## 3. Human Data Application 

In this section, actual fNIRS data from human subjects were used to validate the proposed method. The actual fNIRS data were obtained in the authors’ previous study, but only resting state data were used [2]. In the first subsection, the fNIRS data acquisition is briefly described. The second subsection describes the results of the application of the proposed method.

### 3.1. fNIRS Data Acquisition

Resting state data with a data length of 10 min were selected from ten healthy subjects. The selected subjects are five males and five females (age: 68 ± 5.95 years). Prior to the experiment, each subject was fully informed about the purpose of the study. Written informed consent was obtained from each subject. The entire experiment was approved by the ethics committee of Pusan National University Yangsan Hospital (Institutional Review Board approval number: PNUYH-03-2018-003). 

Hemodynamic responses in PFC were measured with a portable fNIRS device (NIRSIT; OBELAB, Seoul, Republic of Korea) equipped with 24 sources (laser diode) and 32 detectors (a total of 204 channels, including short channel separation) at a sampling rate of 8.138 Hz. NIRSIT uses two wavelengths of near-infrared light (780 nm and 850 nm) to measure concentration changes of HbO and HbR. Only 48 channels with 3 cm of channel distance out of 204 channels were used for this study.

### 3.2. Human Data Analysis

The prediction results for the actual HbO data are shown in Figure 10. Unlike the synthetic data, the amplitude of the ninth wavelet was significantly lower than the other wavelets. A spike appeared in all the wavelets at a particular time, presumably a motion artifact. The MODWT results differed at both ends of the wavelets for the 570 s data and the 600 s data. 

Table 2 shows the results of calculating the mean and standard deviation of the MAEs and RMSEs of the predictions on the real HbO data. Among them, the average value is plotted for easy comparison (Figure 11). The ninth wavelet had the slightest error but the most significant standard deviation across all cases. The fifth and sixth wavelets showed increasingly significant errors until 3 s and 5 s, respectively, then decreased. The seventh and eighth wavelets had more significant errors as the time window increased.

## 4. Discussion

In fNIRS studies, cognitive tasks are used to evaluate cognitive abilities such as working memory, conflict processing, language processing, emotional processing, and memory encoding and retrieval [54]. For example, *N*-back, Stroop, and verbal fluency tasks evaluate working memory, conflict processing, language processing, etc. Such cognitive tasks are also often used to detect brain diseases such as schizophrenia, depression, cognitive impairment, attention-deficit hyperactivity disorder, etc. [55].

Cortical activations caused by cognitive tasks are investigated by a *t*-map, a connectivity map, or extracted features from HbO signals [2,3]. The *t*-map is reconstructed with *t*-values from the GLM method, indicating the dHRF’s weight at each channel. The connectivity map is an image map of correlation coefficients between two channels, which reflects how those two channels are interrelated. Hemodynamic features such as the mean, slope, and peak value have also been used to diagnose brain diseases. Cognitive task analysis can identify activated/deactivated regions and differences between healthy and non-healthy people.

The proposed method was validated in two ways: (i) By comparing synthetic data with and without dHRF, and (ii) by predicting the resting state data. In the synthetic data, the proposed method showed statistically significant differences in the prediction errors between with and w/o dHRF. The prediction errors in human resting state data also showed concordance with the results of synthetic data without dHRF. The agreement between the synthetic data without dHRF and the human resting state data demonstrates that the task-related response can also be differentiated from the proposed method.

Since the hemodynamic signal in this study consisted of 20 s of task and 20 s of rest and had a frequency of 0.025 Hz, it was expected that the eighth wavelet would show a significant difference with and without dHRF. As shown in Figure 6, the wavelet decomposition of the signal with dHRF was different from the signal without in the sixth through ninth wavelets. As expected, a statistically significant difference was found in the eighth wavelet, but the sixth, seventh, and ninth wavelets also showed significant differences. This is likely due to the decomposition of the dHRF into multiple levels when performing the MODWT. 

The LSTM results show that the difference between with and without dHRF is more pronounced when the number of training data points increases. (Figure 7a,b). In addition, the smaller the number of training data points, the smaller the prediction error of the signal with dHRF and the larger the prediction error of the signal without dHRF. This is not surprising, since sufficient data is required for practical training of the LSTM. 

To investigate whether the occurrence of hemodynamic signals can be predicted early, MAEs and RMSEs were estimated by dividing the predicted data into 1 s, 3 s, 5 s, 10 s, 15 s, and 30 s, and the difference in error between the seventh wavelet with and without dHRF was significant early. The difference between the eighth wavelet with and without dHRF was significant at 15 s because it took more than 10 s for the dHRF to rise to the maximum, since it takes time for the dHRF to rise. 

When the proposed method was applied to real data, the error was similar to that of the synthetic data without dHRF. The lowest error occurred in the ninth wavelet, which seems to be due to the lowest signal strength of the ninth wavelet. Initially, wavelets with higher frequencies produced relatively higher errors, but the opposite was true as the prediction time increased. This suggests that as the data length varies, the results of the MODWT change as well, as this is more pronounced at both ends of the data.

Methods to estimate the hemodynamic response and remove noise from fNIRS signals include Kalman filtering [56], Bayesian filtering [57], block averaging [58], general linear models [59], and adaptive filtering [14,60,61]. In addition, initial-dip detection has also been studied for early detection of hemodynamic responses [62,63]. However, these methods rely heavily on the desired hemodynamic function as a reference signal (Table 3). The hemodynamic signal is designed by gamma functions [64], the balloon model [65], the finite element method [66], the state-space method [67,68], etc. These hemodynamic signals are not suitable for use in unknown areas because they depend on the brain region or task being measured. However, the proposed method is differentiated from existing methods in that it does not require a reference signal and can be applied without external devices. 

## 5. Conclusions

The following three implications are made: 

(i) Alleviating the dHRF’s trap: In the conventional methods (i.e., general linear model [59], recursive estimation method [60], etc.), the brain signal is identified by comparing HbO signals with a dHRF. If the correlation coefficient between two signals is high, the measured HbO is attributed to the task. The dHRF computed by convolving a gamma function with the task period contains multiple frequencies, not a single frequency. For example, for a 20 s task followed by a 20 s rest, the dHRF has 0.025 Hz (=1/40 s), and all other components are considered noises. Such multiple frequencies are also seen from the synthetic data analysis, showing that the added 0.025 Hz dHRF affected neighboring frequency bands, see Figure 6. Therefore, if the brain signal is identified with only the dHRF, the neighboring signals are unwillingly included (which could be noises). Hence, the proposed method can alleviate the dHRF’s trap.

(ii) Can handle an unknown task period: In neuroscience, fNIRS has been used to identify brain regions associated with specific tasks and to understand how neural networks function. In particular, regular examinations in daily life are essential for the early detection of cognitive decline due to brain disease or aging. Research on the classification of cognitive decline and brain disease diagnosis using fNIRS is being actively conducted. However, it is challenging to establish classification criteria because hemodynamic signals vary depending on various factors such as age and gender. In particular, it is necessary to compare behavioral data and fNIRS signals for classification, and the duration of cognitive function tests belonging to neuropsychological tests should be pre-designed. Thus, the proposed method can be used when the task period to be observed is unknown or very long.

(iii) Starting time estimation for passive BCI: Recently, passive BCI has become essential for fault-free automotive cars, pilots, etc. In this case, the brain signal’s starting time has to be identified. To estimate the starting time, a moving-window approach can be adopted. If the prediction error becomes large while moving the window, the instance of a significant error can be considered as the starting time of a passive brain signal, and we can generate a BCI command.

The proposed method can overcome the variability in the resting state, which varies from person to person, by predicting the subsequent signal. The predicted signal ought to be removed from the measured signal, and the remaining signal should be analyzed for brain activity. Although the proposed method has some limitations, e.g., large volumes of training data and computation time to train the model for the first time, it is expected to play a significant role in improving the temporal resolution of fNIRS in the future.

## Figures and Tables

**Figure 1 bioengineering-10-00685-f001:**
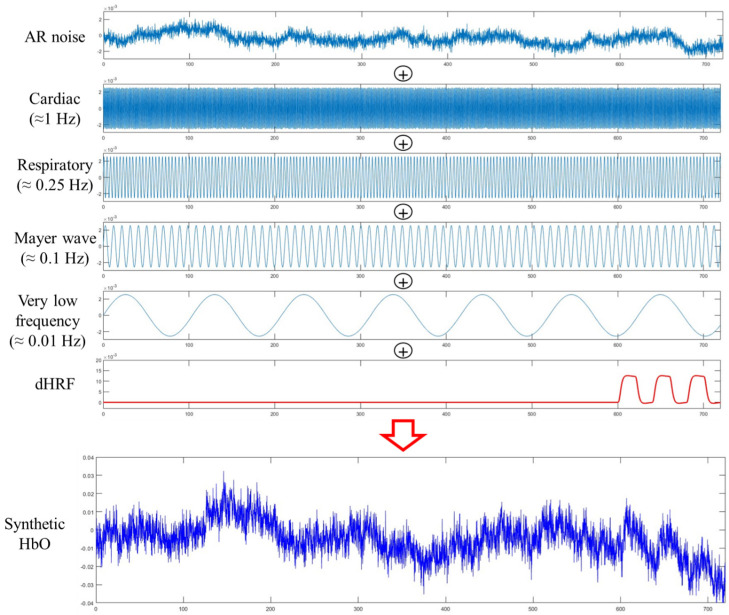
A synthetic HbO signal is made of six components.

**Figure 2 bioengineering-10-00685-f002:**
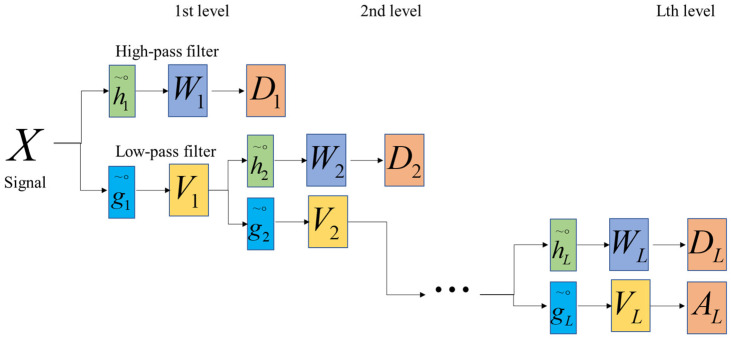
Schematic of the MODWT decomposition.

**Figure 3 bioengineering-10-00685-f003:**
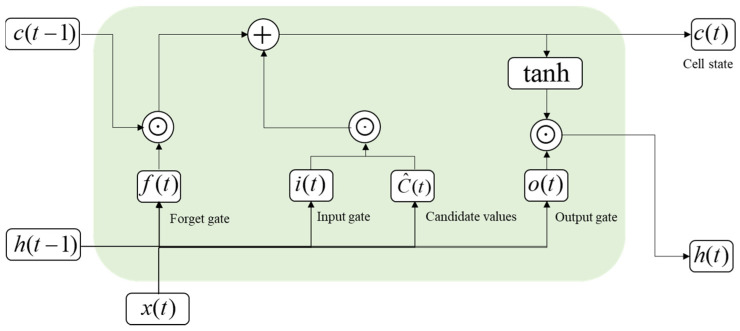
A structure of LSTM layers.

**Figure 4 bioengineering-10-00685-f004:**
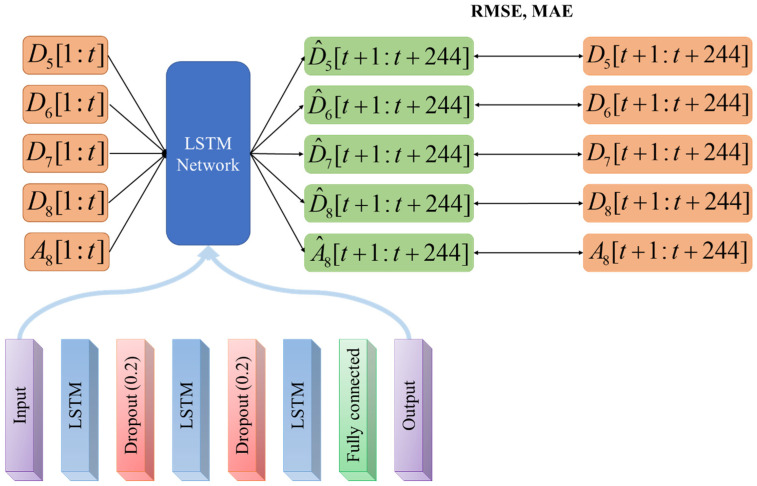
Diagram of the proposed time-series prediction based on MODWT-LSTM.

**Figure 5 bioengineering-10-00685-f005:**
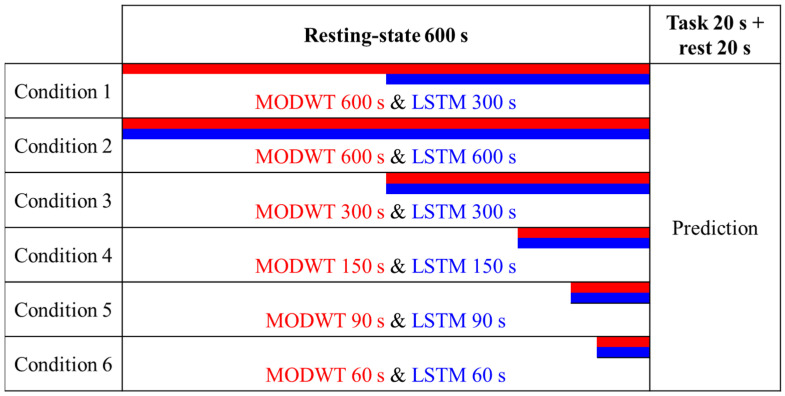
Data segmentation for validation (red: MODWT data length, blue: LSTM data length for training).

**Figure 6 bioengineering-10-00685-f006:**
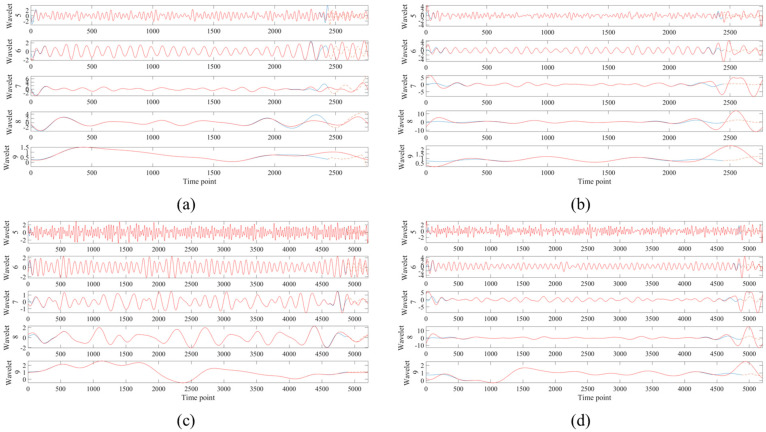
Prediction results from synthetic data: (**a**) Without dHRF (MODWT 300 s, LSTM 300 s), (**b**) with dHRF (MODWF 300 s, LSTM 300 s), (**c**) without dHRF (MODWT 600 s, LSTM 600 s), and (**d**) with dHRF (MODWT 600 s, LSTM 600 s) (blue line: training data, red line: MODWT results including test time, orange dotted line: predicted result).

**Figure 7 bioengineering-10-00685-f007:**
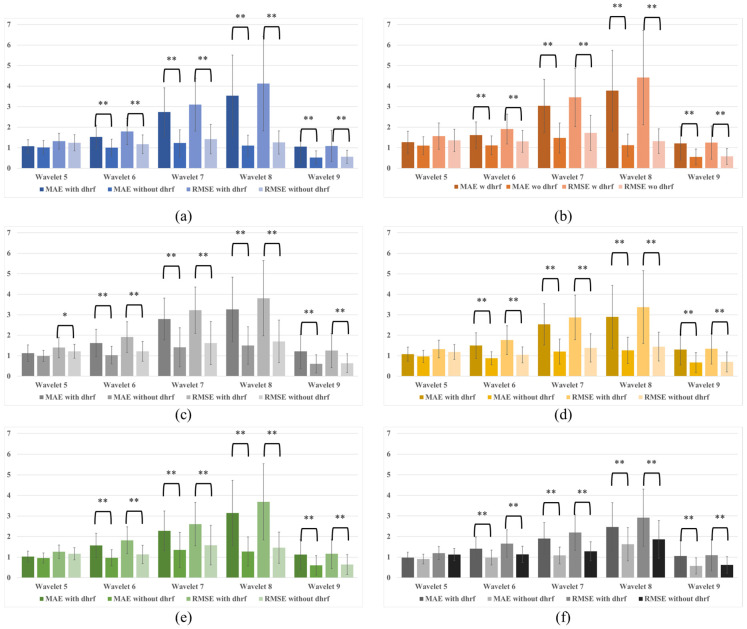
MAEs and RMSEs for wavelets 5–9: (**a**) MODWT 600 s and LSTM training 300 s, (**b**) MODWT 600 s and LSTM training 600 s, (**c**) MODWT 300 s and LSTM training 300 s, (**d**) MODWT 150 s and LSTM training 150 s, (**e**) MODWT 90 s and LSTM training 90 s, and (**f**) MODWT 60 s and LSTM training 60 s (* *p* < 0.05, ** *p* < 0.01).

**Figure 8 bioengineering-10-00685-f008:**
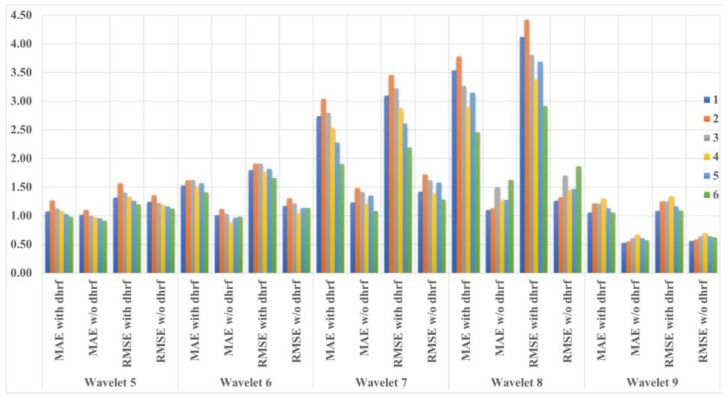
Comparison of MAE and RMSE for all data segmentations: (1) MODWT 600 s and LSTM training 300 s, (2) MODWT 600 s and LSTM training 600 s, (3) MODWT 300 s and LSTM training 300 s, (4) MODWT 150 s and LSTM training 150 s, (5) MODWT 90 s and LSTM training 90 s, and (6) MODWT 60 s and LSTM training 60 s.

**Figure 9 bioengineering-10-00685-f009:**
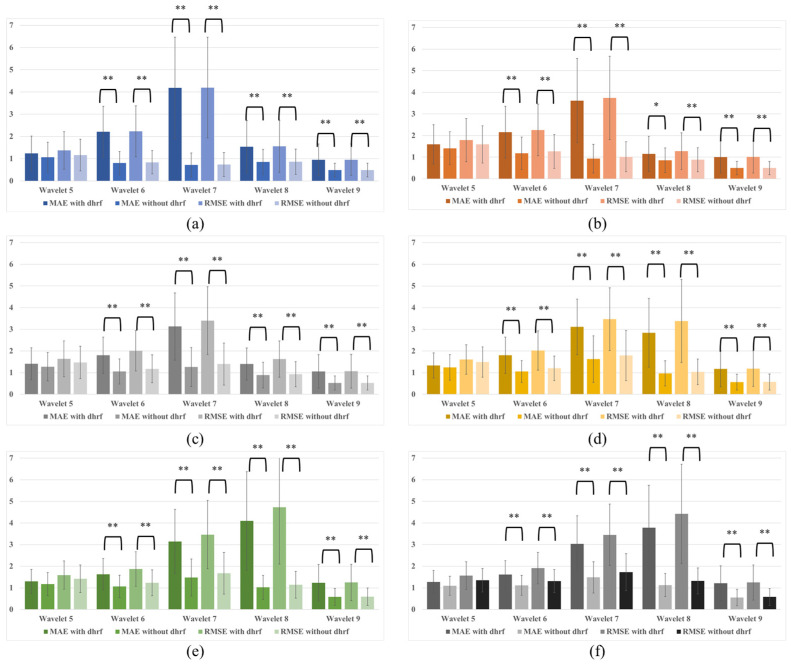
MAEs and RMSEs of 600 s data MODWT-LSTM for each wavelet: (**a**) 1 s, (**b**) 3 s, (**c**) 5 s, (**d**) 10 s, (**e**) 15 s, and (**f**) 30 s (* *p* < 0.05, ** *p* < 0.01).

**Figure 10 bioengineering-10-00685-f010:**
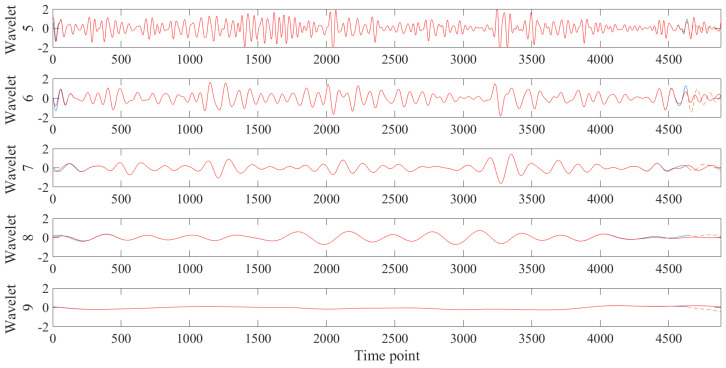
Prediction results of MODWT-LSTM for actual HbO data (blue line: training data, red line: MODWT results including test time points, orange dotted line: predicted result).

**Figure 11 bioengineering-10-00685-f011:**
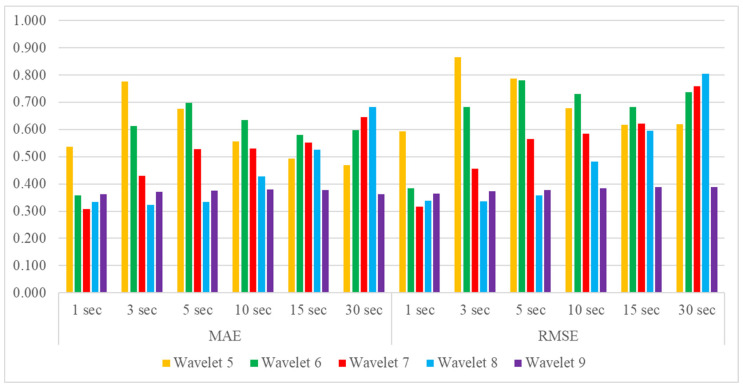
Averaged MAEs and RMSEs of real HbO data MODWT-LSTM for each wavelet by predicted time windows.

**Table 1 bioengineering-10-00685-t001:** Mean and standard deviation of MAEs and RMSEs for synthetic data (* *p* < 0.05, ** *p* < 0.01).

		Wavelet 5				Wavelet 6				Wavelet 7				Wavelet 8				Wavelet 9			
		MAE with dHRF	MAE w/o dHRF	RMSE with dHRF	RMSE w/o dHRF	MAE with dHRF	MAE w/o dHRF	RMSE with dHRF	RMSE w/o dHRF	MAE with dHRF	MAE w/o dHRF	RMSE with dHRF	RMSE w/o dHRF	MAE with dHRF	MAE w/o dHRF	RMSE with dHRF	RMSE w/o dHRF	MAE with dHRF	MAE w/o dHRF	RMSE with dHRF	RMSE w/o dHRF
1	Mean	1.07	1.02	1.32	1.24	1.53	1.01	1.79	1.17	2.74	1.23	3.10	1.42	3.54	1.10	4.12	1.26	1.06	0.53	1.09	0.56
	Std	0.32	0.33	0.38	0.39	0.57	0.41	0.64	0.46	1.18	0.64	1.29	0.72	1.97	0.52	2.30	0.57	0.76	0.32	0.76	0.32
	*p*	0.40		0.35		0.00 **		0.00 **		0.00 **		0.00 **		0.00 **		0.00 **		0.00 **		0.00 **	
2	Mean	1.27	1.10	1.57	1.35	1.62	1.12	1.91	1.31	3.04	1.48	3.45	1.72	3.78	1.13	4.42	1.32	1.22	0.55	1.25	0.58
	Std	0.54	0.44	0.64	0.54	0.64	0.46	0.73	0.53	1.30	0.73	1.42	0.85	1.96	0.54	2.30	0.60	0.81	0.38	0.81	0.39
	*p*	0.09		0.08		0.00 **		0.00 **		0.00 **		0.00 **		0.00 **		0.00 **		0.00 **		0.00 **	
3	Mean	1.13	0.99	1.40	1.22	1.62	1.03	1.91	1.21	2.79	1.41	3.22	1.62	3.27	1.50	3.81	1.70	1.21	0.60	1.25	0.64
	Std	0.41	0.27	0.50	0.34	0.67	0.43	0.75	0.48	1.02	0.96	1.13	1.05	1.58	0.91	1.84	1.04	0.83	0.44	0.83	0.46
	*p*	0.05		0.04 *		0.00 **		0.00 **		0.00 **		0.00 **		0.00 **		0.00 **		0.00 **		0.00 **	
4	Mean	1.08	0.97	1.33	1.19	1.50	0.88	1.76	1.04	2.54	1.21	2.87	1.39	2.90	1.26	3.37	1.44	1.30	0.67	1.34	0.70
	Std	0.35	0.30	0.43	0.37	0.63	0.32	0.71	0.39	1.00	0.61	1.09	0.69	1.53	0.64	1.77	0.70	0.74	0.48	0.74	0.49
	*p*	0.11		0.09		0.00 **		0.00 **		0.00 **		0.00 **		0.00 **		0.00 **		0.00 **		0.00 **	
5	Mean	1.02	0.95	1.26	1.17	1.57	0.97	1.82	1.13	2.28	1.35	2.61	1.58	3.15	1.28	3.69	1.46	1.13	0.61	1.16	0.64
	Std	0.27	0.25	0.32	0.30	0.59	0.39	0.66	0.45	0.97	0.86	1.06	0.96	1.58	0.71	1.85	0.76	0.71	0.47	0.71	0.48
	*p*	0.17		0.13		0.00 **		0.00 **		0.00 **		0.00 **		0.00 **		0.00 **		0.00 **		0.00 **	
6	Mean	0.98	0.91	1.20	1.12	1.41	0.98	1.66	1.13	1.90	1.08	2.19	1.28	2.46	1.63	2.92	1.86	1.05	0.57	1.09	0.62
	Std	0.26	0.24	0.32	0.29	0.57	0.36	0.66	0.40	0.77	0.41	0.86	0.46	1.18	0.81	1.38	0.91	0.77	0.39	0.76	0.40
	*p*	0.19		0.23		0.00 **		0.00 **		0.00 **		0.00 **		0.00 **		0.00 **		0.00 **		0.00 **	

**Table 2 bioengineering-10-00685-t002:** Mean and standard deviation of MAEs and RMSEs for real data.

		MAE						RMSE					
		1 s	3 s	5 s	10 s	15 s	30 s	1 s	3 s	5 s	10 s	15 s	30 s
Wavelet 5	Mean	0.538	0.775	0.677	0.556	0.494	0.469	0.594	0.865	0.786	0.677	0.617	0.619
	Std	0.835	1.056	1.009	0.856	0.796	0.744	0.873	1.138	1.141	0.995	0.932	1.063
Wavelet 6	Mean	0.359	0.612	0.698	0.635	0.580	0.598	0.385	0.682	0.779	0.731	0.682	0.737
	Std	0.643	0.714	0.988	0.905	0.814	0.953	0.669	0.804	1.120	1.056	0.950	1.356
Wavelet 7	Mean	0.309	0.429	0.527	0.529	0.552	0.645	0.316	0.456	0.566	0.585	0.622	0.759
	Std	0.828	1.063	1.297	1.090	1.247	1.520	0.835	1.105	1.356	1.204	1.385	1.802
Wavelet 8	Mean	0.335	0.322	0.334	0.428	0.525	0.683	0.338	0.336	0.358	0.481	0.595	0.805
	Std	1.295	1.172	1.144	1.377	1.680	1.880	1.298	1.208	1.226	1.541	1.852	2.156
Wavelet 9	Mean	0.363	0.372	0.375	0.379	0.378	0.362	0.364	0.374	0.378	0.385	0.388	0.387
	Std	2.157	2.188	2.214	2.225	2.176	1.838	2.157	2.189	2.215	2.227	2.180	1.886

**Table 3 bioengineering-10-00685-t003:** Comparison with the existing methods (adaptive filtering and general linear model) and the proposed method.

	Method	Adaptive Filtering [14]	Bandpass Filter [5]	The Proposed Method
Category				
Low-freq. noise removal capacity		Middle	Low	High
Experiment paradigm (dHRF)		Required	Required	Not required
Processing type		Online	On/offline	Offline
Unknown task period		Cannot handle	Cannot handle	Can handle
Dataset size		Small	Small	Large

## Data Availability

The data and code that support the findings of this study are openly available in https://github.com/sohyeonyoo/MODWT-LSTM.

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
