# Peer review of "Physiological Noise Filtering in Functional Near-Infrared Spectroscopy Signals Using Wavelet Transform and Long-Short Term Memory Networks"

_bioengineering, 2023, doi:10.3390/bioengineering10060685_

Round 1

Reviewer 1 Report

This paper presents a promising new method for filtering physiological noise from fNIRS signals, and has potential applications in a variety of fields. With some minor revisions and additions, it could be a valuable contribution on brain imaging techniques.

-It would benefit from more detailed explanations of how wavelet transform and LSTM networks work, as these may not be familiar to all readers.

-It would be helpful to provide more information about the cognitive tasks used in the study and how the results were analyzed.

-One potential problem with this paper is that it focuses solely on the technical aspects of the proposed method, without discussing the broader implications for neuroscience research. It would be beneficial to include a discussion of how this method could be used to answer specific research questions or advance our understanding of brain function.

-It may be useful to compare the proposed method with existing methods for removing physiological noise from fNIRS signals, in a separate table as “Comparison  Table” to highlight its advantages and limitations.

-Several parts of paper are copied from other works should be modified for example:

Section 3.1 fNIRS data acquisition lines 258-269 is copied from  [R1]

[R1] So-Hyeon Yoo, Seong-Woo Woo, Myung-Jun Shin, Jin A. Yoon, Yong-Il Shin, Keum-Shik Hong. "Diagnosis of Mild Cognitive Impairment Using Cognitive Tasks: A Functional Near-Infrared Spectroscopy Study", Current Alzheimer Research, 2021

-Quality of all figures should be improved and written text should be readable in printed version or 100% zoom.

-Abstract should be improved and should supported by results and numbers.

-Provided conclusion is too long. Maybe it is better, “Discussion & Conclusion” provided as two sections.

Comments on the Quality of English:

The paper is well-written, but there are a few minor grammatical errors that can be corrected.

- In the first sentence of the abstract, "rest state" should be changed to "resting state."

- In the introduction, "novel" should be added before "method" to emphasize that this is a new approach.

-Authors should avoid personal pronouns  (We and I) in the manuscript.

The paper is well-written, but there are a few minor grammatical errors that can be corrected.

- In the first sentence of the abstract, "rest state" should be changed to "resting state."

- In the introduction, "novel" should be added before "method" to emphasize that this is a new approach.

-Authors should avoid personal pronouns  (We and I) in the manuscript.

Author Response

Manuscript ID: 2405322

Title: Physiological noise filtering in functional near-infrared spectroscopy signals using wavelet transform and long-short term memory networks

Authors: S.-H. Yoo, G. Huang, and K.-S. Hong

The authors would like to thank the Editor and two reviewers for their valuable time to review and critique our manuscript. The manuscript has been thoroughly revised upon the reviewers’ constructive comments. The authors’ point-by-point answers to the comments are provided below.

Response to Reviewer #1

Overall Comments: This paper presents a promising new method for filtering physiological noise from fNIRS signals, and has potential applications in a variety of fields. With some minor revisions and additions, it could be a valuable contribution on brain imaging techniques.

       Response: Thank you for the encouraging comment on the potential contribution of our work. The manuscript has been thoroughly revised upon the reviewer’s incisive comments.

Comment 1: It would benefit from more detailed explanations of how wavelet transform and LSTM networks work, as these may not be familiar to all readers.

       Response: The following further explanations of discrete wavelet transforms and LSTM networks were added in the revised version in Sections 2.2 and 2.3, respectively.

  1. i) Discrete wavelet transform (in Section 2.2)

“The discrete wavelet transform (DWT) is a signal processing technique that decomposes a signal into different frequency components at multiple levels of resolution. The DWT works by convolving the signal with a set of filters, called wavelet filters, which capture different frequency bands. The signal is decomposed into approximation and detail coefficients [19], which represent low-frequency components and high-frequency components, respectively. This decomposition is applied recursively to the approximation coefficients to obtain a multi-resolution representation. However, the DWT has several drawbacks, including the introduction of boundary artifacts due to the filtering process, the lack of shift invariance in the decomposition, and the potential loss of fine detail at higher decomposition levels. Therefore, in this work, MODWT was used. The benefits of MODWT over DWT are stated in Section 2.2.”

  1. ii) LSTM network (in Section 2.3)

“LSTM is a type of RNN architecture that addresses the vanishing gradient problem and allows for capturing long-term dependencies in sequential data. LSTM consists of memory cells that store and update information over time. The primary function of an LSTM is to use memory cells that can hold information for long periods. Memory cells can selectively forget or remember information based on input data and past states. This allows the network to learn and remember important information while ignoring irrelevant or redundant information. An LSTM network has three gates (input gate, forget gate, and output gate) that control the flow of information into and out of the memory cells.”

       “Zhang et al. (2018) [51] utilized the DWT in forecasting vehicle emissions and specifically compared four cases: The autoregressive integrated moving average (ARIMA) model, LSTM, DWT-ARIMA, and DWT-LSTM. They reported that adopting DWT improved the performance overall. Individually, between ARIMA and LSTM, LSTM performed better; between ARIMA and DWT-ARIMA, DWT-ARIMA generated improved results; between LSTM and DWT-LSTM, DWT-LSTM was superior; and between DWT-LSTM and DWT-ARIMA, DWT-LSTM demonstrated the best forecasting.”

Comment 2: It would be helpful to provide more information about the cognitive tasks used in the study and how the results were analyzed.

       Response: In compliance with the reviewer’s thoughtful comment, the following three paragraphs were added at the beginning of Section 4, Discussion.

       In fNIRS studies, cognitive tasks are used to evaluate cognitive abilities such as working memory, conflict processing, language processing, emotional processing, and memory encoding and retrieval [54]. For example, N-back, Stroop, and verbal fluency tasks evaluate working memory, conflict processing, language processing, etc. Also, such cognitive tasks are often used to detect brain diseases such as schizophrenia, depression, cognitive impairment, attention-deficit hyperactivity disorder, etc. [55].

       Cortical activations caused by cognitive tasks are investigated by a t-map, a connectivity map, or extracted features from HbO signals [2, 3]. The t-map is reconstructed with t-values from the GLM method, indicating the dHRF’s weight at each channel. The connectivity map is an image map of correlation coefficients between two channels, which reflects how those two channels are interrelated. Hemodynamic features such as the mean, slope, and peak value have also been used to diagnose brain diseases. Cognitive task Analysis can identify activated/deactivated regions and differences between healthy and non-healthy people.

       The proposed method was validated in two ways: i) By comparing synthetic data with and without dHRF, and ii) by predicting the resting state data. In the synthetic data, the proposed method showed statistically significant differences in the prediction errors between with and w/o dHRF. The prediction errors in human resting state data also showed concordance with the results of synthetic data without dHRF. The agreement between the synthetic data without dHRF and the human resting state data demonstrates that the task-related response can also be differentiated from the proposed method.

Comment 3: One potential problem with this paper is that it focuses solely on the technical aspects of the proposed method, without discussing the broader implications for neuroscience research. It would be beneficial to include a discussion of how this method could be used to answer specific research questions or advance our understanding of brain function.

       Response: Upon the reviewer’s critical comments above, the following three implications have been stated in the Conclusion section: i) dHRF’s trap, ii) uncertain task period, and iii) estimation of the starting time for passive BCI.

  1. i) Alleviating the dHRF’s trap: In the conventional methods (i.e., general linear model [59], recursive estimation method [60], etc.), the brain signal is identified by comparing HbO signals with a dHRF. If the correlation coefficient between two signals is high, the measured HbO is attributed to the task. The dHRF computed by convolving a gamma function with the task period contains multiple frequencies, not a single frequency. For example, for a 20 s task followed by a 20 s rest, the dHRF has 0.025 Hz (=1/40 s), and all other components are considered noises. Such multiple frequencies are also seen from the synthetic data analysis, showing that the added 0.025 Hz dHRF affected neighboring frequency bands, see Figure. 6. Therefore, if the brain signal is identified with only the dHRF, the neighboring signals are unwillingly included (which could be noises). Hence, the proposed method can alleviate the dHRF’s trap.

  1. ii) Can handle an unknown task period: In neuroscience, fNIRS has been used to identify brain regions associated with specific tasks and to understand how neural networks function. In particular, regular examinations in daily life are essential for the early detection of cognitive decline due to brain disease or aging. Research on the classification of cognitive decline and brain disease diagnosis using fNIRS is being actively conducted. However, it is challenging to establish classification criteria because hemodynamic signals vary depending on various factors such as age and gender. In particular, it is necessary to compare behavioral data and fNIRS signals for classification, and the duration of cognitive function tests belonging to neuropsychological tests should be pre-designed. Thus, the proposed method can be used when the task period to be observed is unknown or very long.

       iii) Starting time estimation for passive BCI: Recently, passive BCI has become essential for fault-free automotive cars, pilots, etc. In this case, the brain signal’s starting time has to be identified. To estimate the starting time, a moving window approach can be adopted. If the prediction error becomes large while moving the window, the instance of a significant error can be considered as the starting time of a passive brain signal, and we can generate a BCI command.

Comment 4: It may be useful to compare the proposed method with existing methods for removing physiological noise from fNIRS signals, in a separate table as “Comparison  Table” to highlight its advantages and limitations.

       Response: The following comparison table has been added in the revised manuscript, see Section 4. Thank you!

Table 3. Comparison with the existing noise removal algorithms.

Method

Category

Adaptive filtering [14]

Bandpass filter [5]

The proposed method

Low-freq. noise removal capacity

Middle

Low

High

Experiment paradigm (dHRF)

Required

Required

Not required

Processing type

Online

On/offline

Offline

Unknown task period

Cannot handle

Cannot handle

Can handle

Dataset size

Small

Small

Large

Comment 5: Several parts of paper are copied from other works should be modified for example: Section 3.1 fNIRS data acquisition lines 258-269 is copied from  [R1]

[R1] So-Hyeon Yoo, Seong-Woo Woo, Myung-Jun Shin, Jin A. Yoon, Yong-Il Shin, Keum-Shik Hong. "Diagnosis of Mild Cognitive Impairment Using Cognitive Tasks: A Functional Near-Infrared Spectroscopy Study", Current Alzheimer Research, 2021

       Response: Thank you for the keen review: Those parts have been revised or rephrased. Please see the blue parts in the revised manuscript.

Comment 6: Quality of all figures should be improved and written text should be readable in printed version or 100% zoom.

       Response: Upon the reviewer’s comments, the entire figures were re-checked and revised, if necessary.

Comment 7: Abstract should be improved and should supported by results and numbers.

       Response: The abstract section has been revised: The following part has been added.

Abstract: …In the eighth wavelet, the prediction error difference between with- and without-dHRF from the 15-s prediction window appeared to be the largest. On the other hand, in the seventh wavelet, the largest error difference occurred from the 1-s prediction window.

Comment 8: Provided conclusion is too long. Maybe it is better, “Discussion & Conclusion” provided as two sections.

       Response: “Discussion & Conclusion” is divided into two sections: Section 4. Discussion and Section 5. Conclusion.

Comments on the Quality of English:

The paper is well-written, but there are a few minor grammatical errors that can be corrected.

- In the first sentence of the abstract, "rest state" should be changed to "resting state."

- In the introduction, "novel" should be added before "method" to emphasize that this is a new approach.

-Authors should avoid personal pronouns (We and I) in the manuscript.

       Response: A native speaker has proofread the manuscript’s English. All the raised issues were corrected in the revised manuscript.

  • The resting state, instead of the rest state, has been used.
  • The word novel (or innovative) has been added.
  • The sentences with personal pronouns (we, I) have been rewritten.

Thank you very much!

Reviewer 2 Report

The topic of the paper is interesting, and the English language is understandable. There are some comments which are required to be applied as follows:

-At the end of the introduction, please add a paragraph that explain what the next sections are.

-please do not leave section or subsections empty. For example, section ‘’2. Method development’’ is empty and jump to subsection ‘’2.1. Synthetic fNIRS data generation’’. Please explain what the readers are going to be read. The same problem with section ‘’3.’’ and ‘’3.1’’

-The paper has three parts of investigations, one of them is employing the using the deep learning for biosignals. it is expected to add a relative recent paper such as

Hekmatmanesh A, Azni HM, Wu H, Afsharchi M, Li M, Handroos H. Imaginary control of a mobile vehicle using deep learning algorithm: A brain computer interface study. IEEE Access. 2021 Nov 16;10:20043-52.

-Figure 6 and 7 include 6, the labels are difficult to read, please regenerate the plots.

- please explain why ‘’ sym4’’ mother wavelet is selected. Also, please mention why eight levels of decomposition selected.

-it is not professional to write the ‘’4. Discussion & Conclusion‘’ in one section. please prepare it in two separate sections, then it makes possible to find out which one is the analysis, and which one is the conclusion.

The English is understandable and minor language polishing is required.

Author Response

Manuscript ID: 2405322

Title: Physiological noise filtering in functional near-infrared spectroscopy signals using wavelet transform and long-short term memory networks

Authors: S.-H. Yoo, G. Huang, and K.-S. Hong

The authors would like to thank the Editor and two reviewers for their valuable time to review and critique our manuscript. The manuscript has been thoroughly revised upon the reviewers’ constructive comments. The authors’ point-by-point answers to the comments are provided below.

Response to Reviewer #2

Overall Comment: The topic of the paper is interesting, and the English language is understandable. There are some comments which are required to be applied as follows:

       Response: Thank you for the considerate comment. The manuscript was thoroughly revised upon the reviewer’s comments.

Comment 1: At the end of the introduction, please add a paragraph that explain what the next sections are.

       Response: A new paragraph regarding section composition has been added; see the blue part at the end of Section 1.

Comment 2: please do not leave section or subsections empty. For example, section ‘’2. Method development’’ is empty and jump to subsection ‘’2.1. Synthetic fNIRS data generation’’. Please explain what the readers are going to be read. The same problem with section ‘’3.’’ and ‘’3.1’’

       Response: Thank you for the mindful comment. A new paragraph has been added in the empty part between a section title and a subsection title.

 Comment 3: The paper has three parts of investigations, one of them is employing the using the deep learning for biosignals. it is expected to add a relative recent paper such as Hekmatmanesh A, Azni HM, Wu H, Afsharchi M, Li M, Handroos H. Imaginary control of a mobile vehicle using deep learning algorithm: A brain computer interface study. IEEE Access. 2021 Nov 16;10:20043-52.

       Response: Thanks for pointing out the relevant paper. This EEG approach using deep learning has been cited in the text and added as a reference [35] in the revised version.

Comment 4: Figure 6 and 7 include 6, the labels are difficult to read, please regenerate the plots.

       Response: The quality of the figures has been improved in the revised version.

Comment 5: please explain why ‘’ sym4’’ mother wavelet is selected. Also, please mention why eight levels of decomposition selected.

       Response: Sym4 was selected as the mother wavelet because it resembles the canonical hemodynamic response function. Let the number of data be N. Then, the maximum decomposition level becomes less than log2(N). Considering our case’s shortest resting state of 60 s, the data size is 60 s × 8.13 Hz = 487.8. Therefore, the decomposition level in our work was selected by 8, which is the largest integer less than log2(487.8).

Comment 6: it is not professional to write the ‘’4. Discussion & Conclusion‘’ in one section. please prepare it in two separate sections, then it makes possible to find out which one is the analysis, and which one is the conclusion.

       Response: “Discussion & Conclusion” has been divided into two separate sections.  Please see the revised version.

Thank you very much!

Round 2

Reviewer 1 Report

The authors have addressed most of my concerns.

Minor editing of English language required.

Reviewer 2 Report

I am satisfied with the applied answers to the comments.